# Invasive Streptococcal Infection Can Lead to the Generation of Cross-Strain Opsonic Antibodies

Therese de Neergaard,[a] Anna Bläckberg,[a,b] Hanna Ivarsson,[a] Sofia Thomasson,[a] Vibha Kumra Ahnlide,[a] Sounak Chowdhury,[a] Hamed Khakzad,[c] Wael Bahnan,[a] Johan Malmström,[a] Magnus Rasmussen,[a,b] Pontus Nordenfelt[a]

aDivision of Infection Medicine, Department of Clinical Sciences Lund, Faculty of Medicine, Lund University, Lund, Sweden
bSkåne University Hospital, Department of Infectious Diseases, Lund, Sweden
cLaboratory of Protein Design and Immunoengineering, STI, EPFL, Lausanne, Switzerland

**ABSTRACT** The human pathogen *Streptococcus pyogenes* causes substantial morbidity and mortality. It is unclear if antibodies developed after infections with this pathogen are opsonic and if they are strain specific or more broadly protective. Here, we quantified the opsonic-antibody response following invasive *S. pyogenes* infection. Four patients with *S. pyogenes* bacteremia between 2018 and 2020 at Skåne University Hospital in Lund, Sweden, were prospectively enrolled. Acute- and convalescent-phase sera were obtained, and the *S. pyogenes* isolates were genome sequenced (*emm*118, *emm*85, and two *emm*1 isolates). Quantitative antibody binding and phagocytosis assays were used to evaluate isolate-dependent opsonic antibody function in response to infection. Antibody binding increased modestly against the infecting isolate and across *emm* types in convalescent-compared to acute-phase sera for all patients. For two patients, phagocytosis increased in convalescent-phase serum both for the infecting isolate and across types. The increase was only across types for one patient, and one had no improvement. No correlation to the clinical outcomes was observed. Invasive *S. pyogenes* infections result in a modestly increased antibody binding with differential opsonic capacity, both nonfunctional binding and broadly opsonic binding across types. These findings question the dogma that an invasive infection should lead to a strong type-specific antibody increase rather than a more modest but broadly reactive response, as seen in these patients. Furthermore, our results indicate that an increase in antibody titers might not be indicative of an opsonic response and highlight the importance of evaluating antibody function in *S. pyogenes* infections.

**IMPORTANCE** The bacterium *Streptococcus pyogenes* is a common cause of both mild and severe human diseases resulting in substantial morbidity and mortality each year. No vaccines are available, and our understanding of the antibody response to this human pathogen is still incomplete. Here, we carefully analyzed the opsonic antibody response following invasive infection in four patients. Unexpectedly, the patients did not always generate opsonic antibodies against the specific infecting strain. Instead, we found that some patients could generate cross-opsonic antibodies, leading to phagocytosis of bacteria across strains. The emergence of cross-opsonic antibodies is likely important for long-term immunity against *S. pyogenes*. Our findings question the dogma that mostly strain-specific immunity is developed after infection and add to our overall understanding of how immunity to *S. pyogenes* can evolve.

**KEYWORDS** *Streptococcus pyogenes*, *emm* type, antibody response, phagocytosis, bacteremia, antibody function, antigen specificity, group A *streptococcus*, invasive microorganisms, opsonization, sepsis

*S*treptococcus pyogenes, a Gram-positive beta-hemolytic group A *streptococcus* (GAS), is an important human pathogen causing substantial morbidity and mortality at all ages worldwide. *S. pyogenes* infections range from mild to severe systemic

**Ad Hoc Peer Reviewer** Helena Bergsten

Address correspondence to Therese de Neergaard, therese.de_neergaard@med.lu.se, or Pontus Nordenfelt, pontus.nordenfelt@med.lu.se.

The authors declare no conflict of interest.

ones and result in a variety of diseases, such as erysipelas, tonsillitis, bacteriemia, and rheumatic fever. Each year it is estimated to cause more than 700 million mild skin and throat infections and around 600,000 invasive life-threatening ones, such as sepsis and necrotizing fasciitis (1). Over the years, *S. pyogenes* has remained sensitive to penicillin, but reduced penicillin sensitivity in a few isolates has been recently reported (2, 3). Furthermore, despite great efforts, no vaccines have been developed (4). Therefore, it is important to further understand the immune response to this pathogen.

A crucial part of the defense against pathogens is opsonizing antibodies, which, when bound to the pathogen, enhance its eradication by phagocytosis. However, *S. pyogenes* has evolved multiple strategies to resist phagocytosis (5–7). A major virulence factor in this process is the streptococcal M protein, encoded by the *emm* gene. M protein can reverse antibody orientation through Fc binding (8, 9) and interact with multiple antiphagocytic proteins (10, 11), and it exhibits antigenic diversity through its hypervariable region, resulting in >250 *emm* types (4). The M protein covers most of the bacterial surface and is an important target for the immune system through type-specific antibodies. These antibodies start to appear around 4 weeks after a GAS infection (12) and persist for up to 30 years (13, 14). It is generally believed that immunity is *emm* type specific and is acquired through the development of protective type-specific antibodies. Initial studies suggested that only antibodies against the hypervariable part of the M protein were opsonic (15), but later studies reported that antibodies to conserved binding sites can also be opsonic (16–18). These findings indicate the presence of anti-M antibodies, which may convey immunity to more than one *emm* type, providing broader protection. The immune response might also target other parts of the bacteria, such as carbohydrates of the cell wall (19), which could convey more general cross-type immunity. Children and adolescents suffer from recurring GAS infections; however, these infections decrease radically in adulthood. Therefore, it is suggested that through those repeated exposures over time, a broader and more long-term immunity is developed (20).

Specific antibodies do not always activate immune functions, and antibody responses can be described as opsonic or nonopsonic. In contrast to its opsonic counterpart, a nonopsonic antibody binds to its antigen without contributing to eradicating the pathogen by phagocytosis (16, 21, 22). Recently, Bläckberg et al. reported that patients suffering from an invasive *Streptococcus dysgalactiae* infection failed to develop protective opsonic antibodies (22), and Uddén et al. found that the generation of nonopsonic antibody responses was correlated with the invasive nature of the *Streptococcus pneumoniae* infection (23). However, it is unknown to what extent this occurs for *S. pyogenes* infections and how important the nature of the infection, and in particular invasive disease, is in developing opsonizing antibodies to *S. pyogenes*.

To better understand the functional immune response in *S. pyogenes* infections, we assessed antibody binding and opsonic capacity in four patients during and after an invasive *S. pyogenes* infection. Interestingly, we report the development of both nonopsonic type-specific antibodies and broad opsonic antibody responses across different *emm* types in these patients.

## RESULTS

**Clinical characterizations of patients and bacterial isolates.** Four patients (patients A to D [PA to PD]) with *S. pyogenes* bacteremia were enrolled. Their clinical characteristics are presented in Fig. 1A and Table 1. In summary, two females (PB and PC) were infected with isolates of type *emm1*, and two males had *emm118* (PA) or *emm85* (PD). The primary infection foci were skin and soft tissue. Upon admission, all patients were hemodynamically stable, but within 48 h, patient D acquired septic shock (SEPSIS-3 criteria [24]). Patient D's symptoms commenced 2 weeks before admission, in contrast to the others, who were admitted to the hospital almost immediately, suggesting that the serum from this patient could have been in a different immune response phase than the other acute-phase samples. The patients' immunoglobulin (IgA, IgG, and IgM) concentration in serum changed between admission (acute) and 6

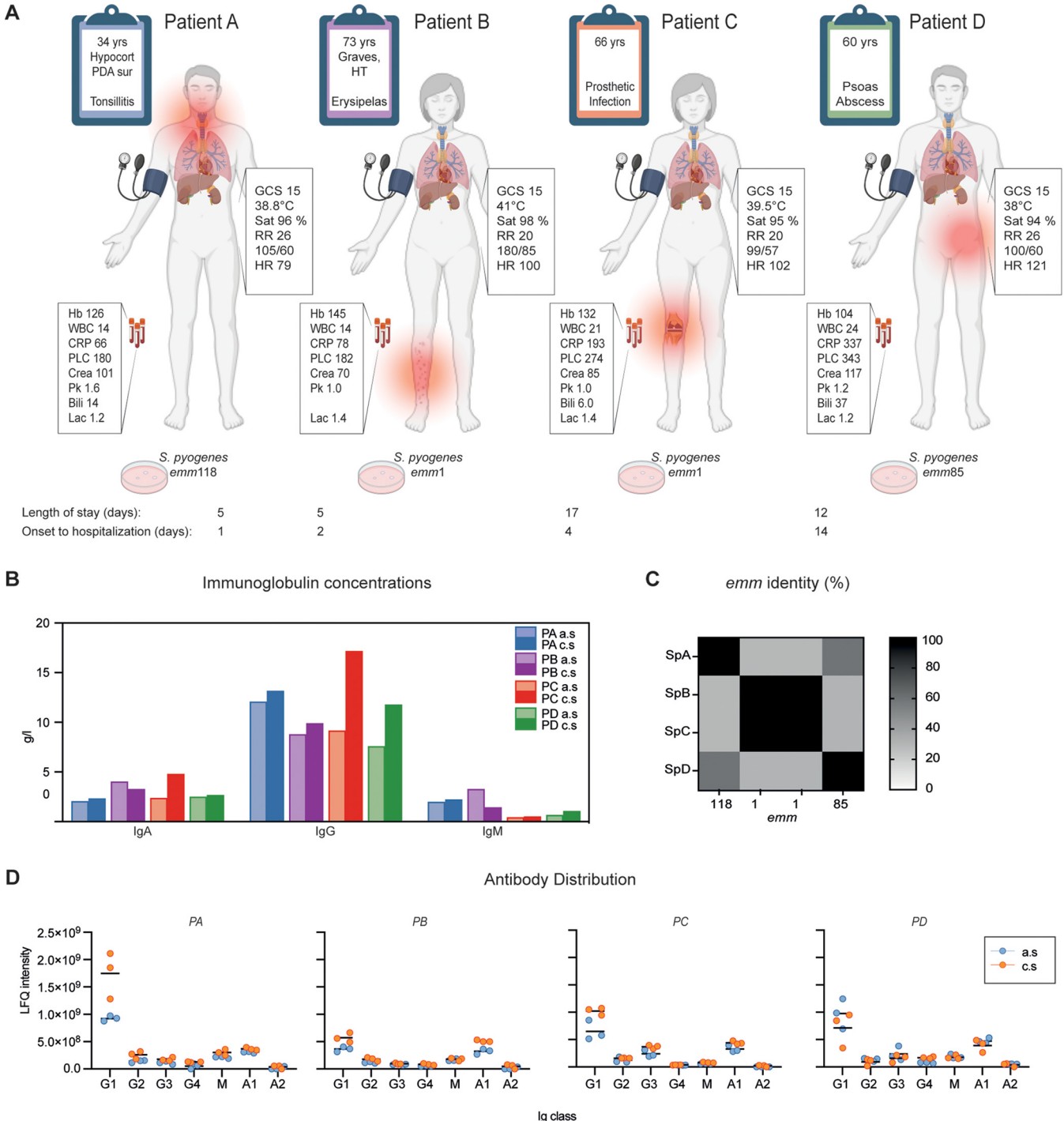

**FIG 1** Characteristics of the patients and their infecting isolates. (A) The four patients (PA to PD) with invasive *S. pyogenes* infection included in the study. Parameters given are as follows: GCS, Glasgow coma scale; RR, respiratory rate; Sat, saturation; blood pressure (in millimeters of Hg); HR, heart rate; Hb, hemoglobin (in grams per liter); WBC, white blood cells count ($10^9$ per liter); CRP, C-reactive protein (in milligrams per liter); PLC, platelet count ($10^9$ per liter); Crea, creatinine (in micromoles per liter); PK(INR), prothrombin complex international normalized ratio; Lac, lactate (in millimoles per liter). (B) Immunoglobulin concentrations in acute-phase serum (a.s) and convalescent-phase serum (c.s), as reported from the clinical diagnostic data. (C) Multiple-sequence alignment and percent identity of the four detected M protein sequences, including two identical *emm*1 sequences from SpB and SpC, the *emm*85 sequence from SpD, and the *emm*118 sequence from SpA. (D) Immunoglobulin distribution was determined by mass spectrometry for each patient's serum. The line represents the mean (*n* = 3). Panel A was created with BioRender.com.

weeks later (convalescent), with the largest difference being an increase in IgG for patients C and D (PA, 9%; PB, 13%; PC, 87%; PD, 55%) (Fig. 1B). To determine the subclass distribution of the immunoglobulins, we performed a mass-spectrometric analysis of the patients' sera (Fig. 1D). After infection, IgG1 increased for three of the four

**TABLE 1** Clinical features of patients with invasive *S. pyogenes* infection

| Patient | Sex and age (yrs) | *emm* type | Underlying disease[a] | No. of days: | | Focus of infection |
|---------|-------------------|-----------|----------------------|--------------|--|---------------------|
| | | | | From onset to hospitalization | Of hospital stay | |
| A | M, 34 | *emm*118 | Hypocortisolemia, PDA surgery | 1 | 5 | Tonsillitis |
| B | F, 73 | *emm*1 | Graves with thyrotoxicosis, HT | 2 | 5 | Erysipelas |
| C | F, 66 | *emm*1 | Healthy | 4 | 17 | Prosthetic infection |
| D | M, 60 | *emm*85 | Healthy | 14 | 12 | Psoas abscess |

[a]PDA, patent ductus arteriosus; HT, hypertension.

patients, whereas patient D had more IgG1 in the acute-phase sera. Additionally, for patient C, the IgG distribution shifted to a noticeably higher level of IgG3 in convalescent-phase sera. The *S. pyogenes* isolates were sequenced (see the supplemental material), and the M protein sequences were compared (Fig. 1C). The *emm*1 isolates (SpB [*S. pyogenes* from patient B] and SpC) had identical M proteins, but the overall similarity between *emm*1, *emm*85, and *emm*118 was relatively low (30%). However, there was a higher sequence similarity between *emm*85 and *emm*118 when pairwise aligned (59.3%). In summary, our patient group was small, with different ages and disease severity but similar health status. The infecting agents consist of three different *emm* types, with two patients infected by the same type.

**Antibody binding is increased after invasive *S. pyogenes* infection.** To determine the effect an invasive *S. pyogenes* infection has on antibody binding, we opsonized each *S. pyogenes* isolate bacteria with the corresponding paired sera. The nonspecific monoclonal antibody omalizumab (Xolair) was a control for Fc binding, and intravenous immunoglobulin (IVIG) was a positive control. To evaluate the contribution of the M protein as an antigen, we included an M protein-deficient *S. pyogenes* mutant, SF370dM (referred to here as dM). When the whole curve in the convalescent-phase sera was analyzed, all patients had a significant increase in IgG bound to the bacteria compared to the level in acute-phase sera (Fig. 2A), and already at 0.1% serum, differences could be seen, albeit not statistically significant (Fig. 2C). Interestingly, the amounts of IgG bound to dM increased from low at low serum concentrations to almost the same level as those of the clinical isolates when measured at the higher concentrations (Fig. 2B). The result thus indicates the presence of both IgGs with high affinities against M proteins and additional antigens in the absence of M protein. To evaluate the relative change of bound IgGs within each patient's sera, the level of IgG bound to the bacteria in convalescent-phase serum was expressed as a fold change relative to the level of IgG bound in acute-phase serum (Fig. 2D). The M1 patients (PB and PC) had the largest relative change among the isolates, while patients A and D had almost no difference. Nonetheless, the dM strain had the highest relative increase for each patient.

To quantify the affinities of the binding IgG against the pathogen, we analyzed the binding curves using a bacterium-antibody binding model (25) (Fig. 2E; also, see Fig. S1A in the supplemental material). Patient C had the highest increase in affinity after infection. However, both patients A and D had higher affinities than patient C already in acute-phase sera. To summarize, antibody binding was increased to the infecting isolate after an invasive *S. pyogenes* infection regardless of *emm* type.

**Invasive infection leads to a differential opsonic response.** To evaluate the dynamics of phagocytosis, we studied it at low to high bacterium-to-cell ratios (multiplicity of prey [MOP]) using heat-killed bacteria and THP-1 phagocytes as previously described (26). By heat inactivating the sera, we excluded any contribution from the complement system, in order to focus on Fc-mediated phagocytosis. We assessed the phagocytic ability of the phagocyte population based on the portion of phagocytes that can associate with or internalize their prey (Fig. 3A). To evaluate possible differences in phagocytosis between live and heat-killed bacteria, phagocytosis with THP-1 cells was assessed on both live and heat-killed isolates opsonized with their corresponding sera (Fig. S3). No obvious differences in either association or internalization

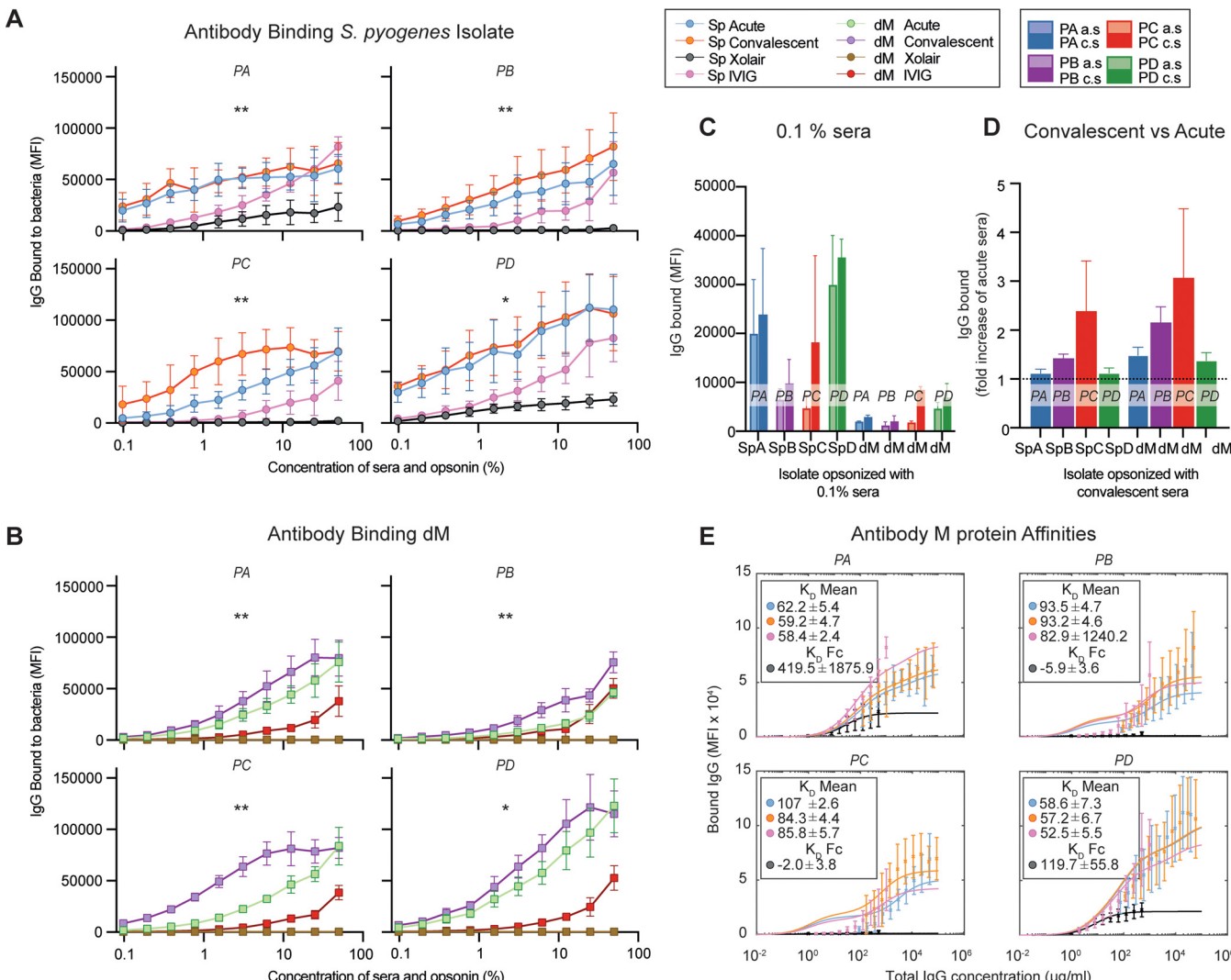

**FIG 2** Antibody binding and distribution after invasive GAS infection. The acute- and convalescent-phase sera from four patients (PA to PD) with *S. pyogenes* (Sp) invasive infection were assessed. (A to D) Each patient isolate (Sp) was coded with the corresponding patient letter (A to D) (SpA, *emm*118; SpB, *emm*1; SpC, *emm*1; SpD, *emm*85). The patient isolates and a strain lacking the M protein (dM) were opsonized with the corresponding sera and controls in serial dilution (see the color key). The undiluted concentration was 2 mg/mL for the polyclonal IgG (IVIG) and 1 mg/mL for monoclonal nonspecific IgG (omalizumab [Xolair]). Binding was determined with IgG Fab-specific far-red fluorescent antibodies. Data were acquired through flow cytometry and are presented as means and standard deviations (SD) ($n = 4$). (A to C) The amount of IgG bound to the isolate (A) or dM (B) for each opsonin expressed as the fluorescence intensity of the secondary antibody. Wilcoxon's matched-paired signed-rank test was performed on the complete binding curves of acute- and convalescent-phase serum for the corresponding isolate and separately for their binding to dM to assess any significant differences between the acute- and convalescent-phase-serum binding. *, $P < 0.05$; **, $P < 0.01$. (C) Data from panels A and B were plotted at a 0.1% serum concentration. (D) Change in binding for convalescent- compared to acute-phase serum, expressed as fold increase over bound IgG in acute-phase serum. The baseline, set at 1, is marked with a line. (E) Affinities of each serum for M protein through modeling of binding data. The affinity for sera and IVIG is designated the $K_D$ mean $\log_{10}$ reference (per nanomolar unit), while for omalizumab, the M-protein Fc affinity was named $K_DFc$ (per nanomolar unit).

were noted between live and heat-killed bacteria. We also compared the performance of THP-1 cells to that of neutrophils isolated from healthy donors (Fig. S4). Neutrophils were better at phagocytosis in general when compared at the same MOP but also exhibited a large donor variation. Otherwise, there was no apparent difference in trends between the THP-1 cells and neutrophils, making THP-1 cells viable as an option to evaluate phagocytosis.

In the convalescent-phase serum, the association and internalization were significantly increased for patients A and C compared to acute-phase sera. On average, 50% more phagocytes were associated with bacteria (Fig. 3B), and 75% more phagocytes had internalized at least one bacterium with patient C convalescent-phase serum (Fig. 3C). The increase for patient A was 3% for association and 25% for internalization.

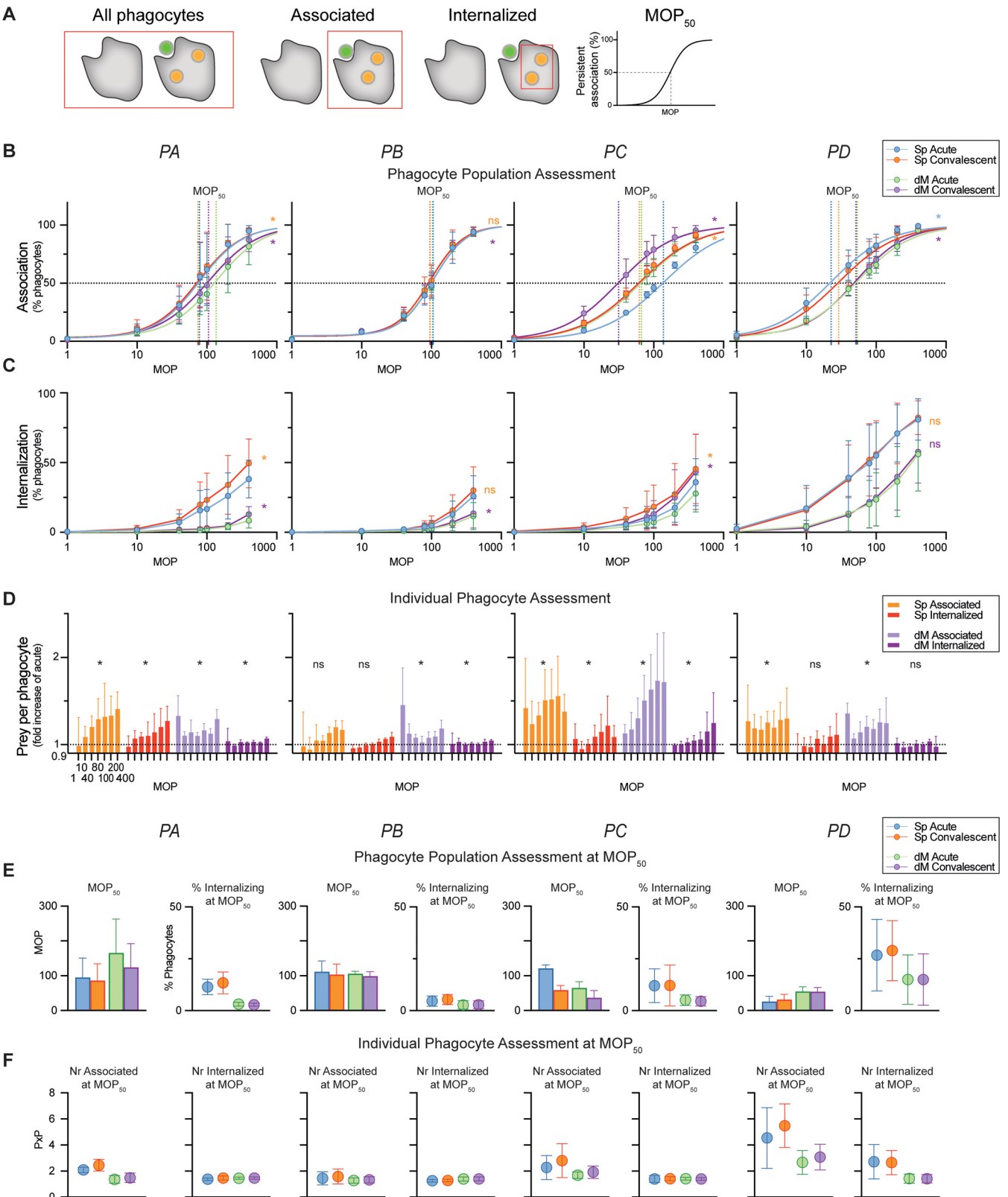

FIG 3 Assessment of the phagocytic response during infection. (A) Schematic of the parameters used for the analysis of phagocytosis. Associated cells are those that have at least one bacterial unit associated with them, whereas internalized cells have at least one bacterial unit inside the cell. $MOP_{50}$ is a measure of the number of bacteria per phagocyte required to reach 50% persistent association. (B to F) The acute- and convalescent-phase sera from four patients (PA to PD) with *S. pyogenes* invasive infection were assessed. THP-1 cells were incubated (30 min; 150 $\mu$L; MOP, 0 to 400; 37°C) with either the infecting isolate (Sp) or a strain lacking M protein (dM) after the bacteria were fluorescently doubled stained with a pH-stable (Oregon green) and a pH-

Patient D's acute-phase serum mediated higher association, whereas there was no difference in internalization compared to convalescent-phase serum. For patient B, there was no significant increase in either association or internalization. For dM, there was a significant increase from acute- to convalescent-phase sera for each patient in association ability and for all except patient D in the internalization ability. This indicates that the increase seen in phagocytosis when comparing convalescent- to acute-phase sera cannot be explained only by antibodies binding to the M protein.

In Fig. 3D, from an analysis of bacterial fluorescence intensities at the single-cell level, we provide an assessment of the phagocytic ability of each phagocyte, meaning its individual ability to adhere to and internalize bacteria. Compared to acute-phase serum as the baseline, an increased association was detected for patients A, C, and D at all MOP ($P < 0.05$; median increases were 29% [PA], 42% [PC], and 25% [PD]) but only at the highest concentration for patient B. Internalization, on the other hand, was significantly improved only for patients A and C (10% and 8.8%, respectively). For all the convalescent-phase sera, dM was significantly more associated with cells than for acute-phase sera.

In Fig. 3E, we show a comparison of the association capacity between the patients' sera in a standardized manner by determining at what MOP 50% of the phagocyte population was associated with bacteria ($MOP_{50}$). Patients A, B, and C sera required similar MOPs, around 100, while patient D serum needed less than half to reach 50% association. Only patient C had a clear improvement in association capacity with convalescent-phase sera (the $MOP_{50}$ was halved). At the $MOP_{50}$, serum from patient D not only had the highest number of bacteria interacting with each phagocyte (Fig. 3F) but also had the highest bacterial internalization in the phagocytes (Fig. 3E). When convalescent-phase serum was compared to acute-phase serum, patients A and D had slightly increased proportions of phagocytes internalizing bacteria (Fig. 3F). For patients A, C, and D, the individual phagocyte adhered to more bacteria in convalescent-phase sera, while internalization was unaffected. There are no differences for patient B and dM samples on the population or individual level.

When the different parameters analyzed were summarized, patient C had the most evident improvement in phagocytosis on both the population and individual-phagocyte levels after infection; patient A had some improvement, while patient B had none. Patient D had the highest opsonic ability overall, which remained in the convalescent-phase serum.

**Infection-induced opsonic antibodies are cross-reactive, while the nonopsonic response seems to be _emm_ type specific.** To determine whether our findings were _emm_ type specific, we evaluated the effects of heat-inactivated sera on binding and phagocytosis across isolates. Overall trends are shown as heat maps (Fig. 4A and C), with quantitative analysis in Fig. 4B and D. The IgG binding was significantly increased in the convalescent-phase sera compared to acute-phase sera across the different isolates, except for SpA (_S. pyogenes_ isolate infecting patient A) opsonized with patient D serum, for which the increase was more modest (Fig. 4A and B). As expected, we saw the highest binding to the infecting isolate with sera from patients A and D but, interestingly, not with sera from patients B and C, which both were infected with _emm_1 strains. In addition, the _emm_1 isolates had fewer antibodies bound, independent of which sera were tested. The _emm_1 isolates were also the least phagocytosed by each patient's serum (Fig. 4C and D). However, there was a significant increase in phagocyte

**FIG 3** Legend (Continued)

sensitive (CypHer-5E) dye and opsonized in 5% of the corresponding sera (see the color key). Data were acquired through flow cytometry and are presented as means and SD ($n = 4$). (B and C) The percentage of phagocytes in the phagocytic population associating with (B) and internalizing (C) opsonized bacteria for each patient. In panel B, the average of the fitted persistent association curves is shown, and 50% association is marked with a line for each curve. The complete phagocytosis curves of acute- and convalescent-phase samples were compared for any significant differences with Wilcoxon's matched-paired signed-rank test. *, $P < 0.05$. (D) The change in the number of bacteria an individual associating phagocyte has associated with and internalized in convalescent- compared to acute-phase serum, expressed as fold increase over the acute-phase-serum value. Wilcoxon's signed-rank test was performed on any significant fold changes. *, $P < 0.05$. The baseline is visualized with a line at 1. The MOP range is 1 to 400. (E) In each pair of graphs, the MOP when 50% of the phagocytic population has become associated ($MOP_{50}$), based on the curves in panel B, is on the left, while the percentage of phagocytes internalizing at $MOP_{50}$ is on the right. (F) In each pair of graphs, the average number of prey units per phagocyte (PxP) that are associated and internalized at $MOP_{50}$ are on the left and right, respectively.

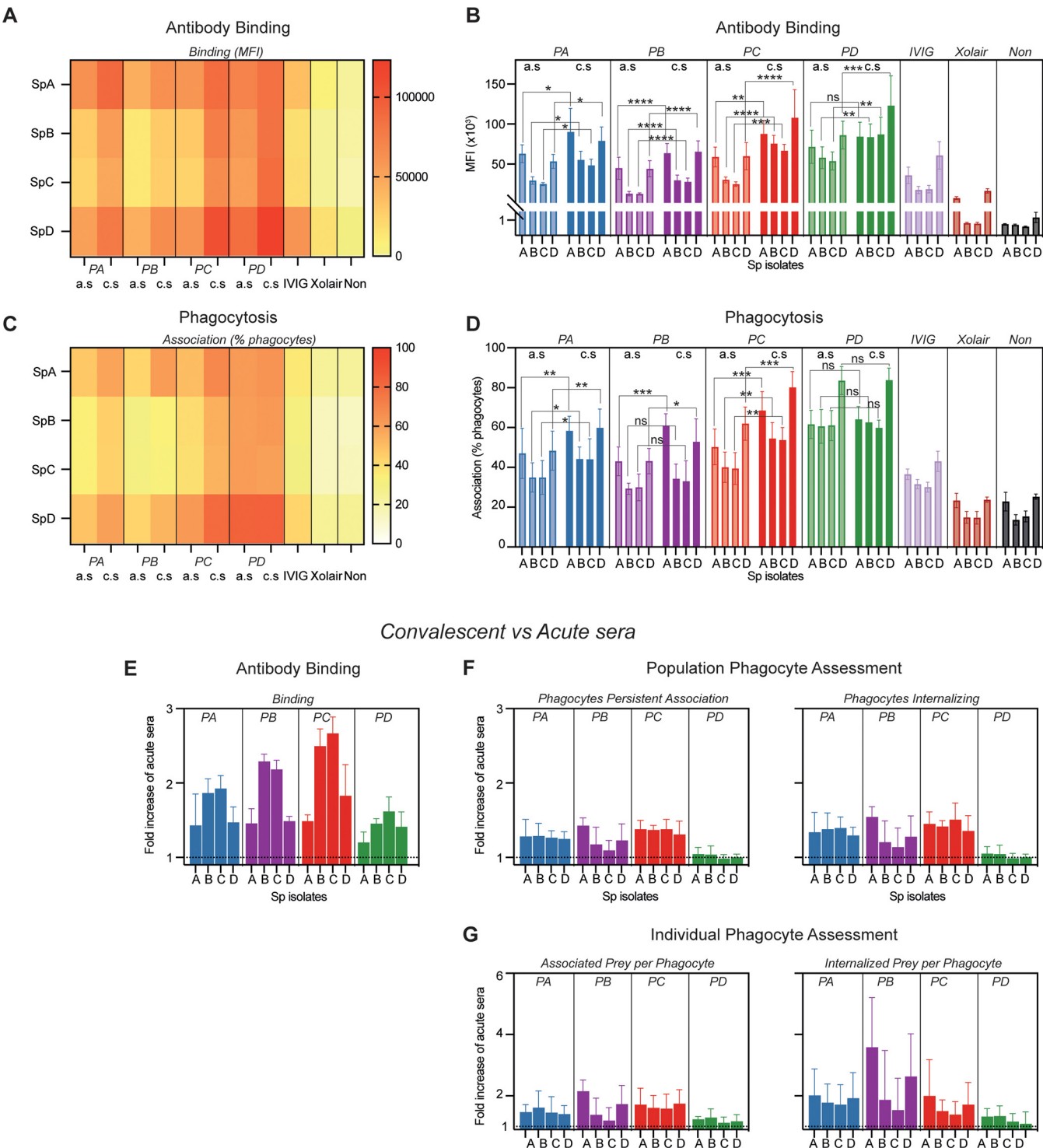

**FIG 4** Antibody binding and opsonic function across strain for each patient. Acute-phase sera (a.s) and convalescent-phase sera (c.s) from four patients (PA to PD) with *S. pyogenes* invasive infection were assessed. The different isolates (Sp) were fluorescently doubled stained with a pH-stable (Oregon green) and a pH-sensitive (CypHer-5E) dye and opsonized in 5% of each serum. The concentration was 0.5 mg/mL for polyclonal IgG (IVIG) and monoclonal nonspecific IgG (omalizumab [Xolair]), and no opsonin was used for the negative control. Binding was determined with IgG Fab-specific far-red fluorescent antibodies. Phagocytosis was performed with THP-1 cells incubated (30 min; 150 μL; MOP, 80; 37°C) with each isolate. Data were acquired through flow cytometry and are presented as means and SD (*n* = 4). (A to D) Heat maps and histograms visualizing IgG bound to each isolate (A and B) and the percentage of phagocytes associating with the bacteria (C and D). Significance was tested with two-way ANOVA and Šidák's multiple-comparison test. *, *P* < 0.05; **, *P* < 0.01; ***, *P* < 0.001; ****, *P* < 0.0001. (E to G) Changes in binding (E) and phagocytosis (F and G) for convalescent- compared to acute-phase serum for each patient, expressed as fold increase over the corresponding acute-phase-serum value. The baseline is visualized with a line at 1. Panel F shows the change in the portion of the phagocyte population that associated and internalized bacteria, while panel G shows the change in the individual phagocytes' capacity to associate (left) and internalize (right) bacteria.

**TABLE 2** Adaptive response after *S. pyogenes* infection[a]

| Patient | Infecting isolate type | *emm* group | Specific isolate | | | | CS cross-reactive[b]: | |
|---|---|---|---|---|---|---|---|---|
| | | | Binding (MFI, $10^3$) | | Phagocytosis (%) | | | |
| | | | AS | CS | AS | CS | Binding | Phagocytosis |
| A | *emm*118 | E | 63 | +27* | 47 | +11** | +26* | +9.23* |
| B | *emm*1 | A–C | 13 | +17**** | 29 | +5.0ns | +19**** | +9.73* |
| C | *emm*1 | A–C | 25 | +42*** | 39 | +14** | +45**** | +18*** |
| D | *emm*85 | D | 86 | +37*** | 84 | +0.2ns | +26** | +1.98ns |

[a]Opsonized in 5% paired sera; phagocytosis (persistent association) at a MOP of 80. AS, acute-phase serum; CS, convalescent-phase serum; MFI, median fluorescent intensity. + indicates increase from acute-phase serum. Two-way ANOVA was used to assess significance. *, $P < 0.05$; **, $P < 0.01$; ***, $P < 0.001$; ****, $P < 0.0001$; ns, not significant.
[b]The median value is reported.

association for all the isolates opsonized in sera from patients A and C. In contrast, only SpA and SpD were improved for patient B, whereas patient D had no significant increase at all. Similar trends were seen with the antibody controls IVIG and omalizumab, but with overall lower levels than with the patient sera.

In Fig. 4E to G, we compare the convalescent-phase sera to the acute-phase sera. The largest increase in the antibody binding was against the *emm*1 isolates (SpB and SpC) for all patients, and patient C had the largest increase in antibody binding for all isolates (Fig. 4E). The response in phagocytosis, at both the population (Fig. 4F) and individual-phagocyte (Fig. 4G) levels, was elevated in the same manner as the phagocyte association (Fig. 4C and D). Thus, after infection, patient B developed antibodies that were opsonic against isolates of other M types but nonopsonic against its infecting *emm*1 isolate (SpB). Nonetheless, serum from patient C, also infected with *emm*1, had increased binding and function against all isolates, including SpB, in convalescence. Hence, opsonic antibodies can be generated after an *emm*1 infection. Serum from patient A, infected by the *emm*118 isolate, increased antibody binding and phagocytosis across the strains, indicating a broad and improved opsonic antibody response after infection. Serum from the *emm*85-infected patient D maintained a high level of phagocytosis across strains, with no further increase but with increased levels of antibody binding after infection (Fig. 4E to G). To test the broadness of these results, phagocytosis was assessed on four other isolates (*emm*4, *emm*5, *emm*12, and *emm*89) opsonized with each patient's sera (Fig. S5). Here, the data indicate that phagocytosis was increased in all of the convalescent-phase sera, not for all isolates but for at least one per patient. To summarize (see also Table 2), after invasive *S. pyogenes* infection, cross-strain opsonic antibodies can be developed. On the other hand, nonopsonic binding antibodies are generated against specific types, and here, it was seen primarily with the *emm*1 type.

## DISCUSSION

The generation of opsonizing antibodies is a vital step in developing pathogen-specific immunity. In the present work, we quantified the opsonic capacity of serum from patients during and after invasive *S. pyogenes* infection. Our results show that a modest increase in antibodies binding to the bacteria occurs after infection. It should be pointed out that this is from a relatively high basal level, as seen when compared to acute-phase samples. However, this increase in binding antibodies does not always lead to an improved functional response in terms of phagocytosis, as seen with patients B and D. These findings are consistent with previous studies on *S. pyogenes* immunity, where antibodies binding to the conserved region of the M protein typically did not result in a bactericidal effect (15, 16). This nonopsonic response did not seem to be specific for *emm*1 type infection, since the other patient (PC) infected with the *S. pyogenes* of the identical type did generate opsonic antibodies, nor was it a lack of complete response, since PB did develop opsonic antibodies toward other *emm* types. The mechanism behind these two different responses is unknown, and it might be difficult to draw firm conclusions from a limited set of patients. Still, we speculate that *S.*

*pyogenes* might have mechanisms influencing the immune system so that it generates nonfunctional antibodies. Our results show that it is important to properly assay both antibody titers and antibody function to characterize an immune response.

We did not find any correlation between the clinical outcome and the different responses. However, it is noticeable that patient D, with the most severe infection, did differ from the others. The opsonic capacity was already high in his acute-phase serum, but since he had the longest delay to hospitalization, his serum samples might have been in a different immune response phase than those from the other patients. He had no improved functional response in terms of phagocytosis in his convalescent-phase serum, which might be a contributing factor to the severity of his disease. However, the development of a severe infection is complex, and factors affecting a clinical outcome are hard to dissect.

Interestingly, the opsonic antibodies generated by the patients were cross-reactive and enhanced phagocytosis across types. Even if patients suffering from *S. pyogenes* invasive infection rarely get reinfected (27), suggesting broader protection, *S. pyogenes* immunity is typically described to be type specific with protective antibodies (15, 28). The immune response to the hypervariable region appears to be subdominant (29), with the majority of the antibodies instead being generated to the more conserved, and what appears to be mostly nonopsonic, C-repeat regions (30). After an invasive infection, patients are expected to have a strong and specific antibody response rather than the modest and cross-reactive response seen in patients studied here. However, during the last decade, studies have reported the development of broadly opsonic antibodies in animal vaccine trials (31) and after superficial skin infections in school children (32). Furthermore, we recently found a protective human-derived antibody with opsonic function across a broad range of *emm* types (16). The latter observation shows that there is a possibility of developing opsonic and protective antibodies to the conserved regions of M protein; however, the chance of that happening remains to be established. Here, we have provided clinical data on the development of cross-type opsonic antibodies after invasive *S. pyogenes* infection, which to our knowledge has not previously been described.

We noticed that while there was only a modest increase in IgG levels in convalescent-phase sera, there was a larger increase in bacterial binding. Hypothetically, this could be explained by the enhancement of preexisting antibodies to *S. pyogenes* by further affinity maturation rather than by the induction of clonal expansion of new antibodies. The baseline binding is also already relatively high even with acute samples compared to that of IVIG at similar antibody concentrations. This likely indicates that the patients have already been exposed to *S. pyogenes* at an earlier point in life and possibly that the immune system, to a certain extent, was already ramping up the production of antibodies at the time of sample collection. Another possibility is that *S. pyogenes* shares proteins or other structures with other species, which would give the immune system a preexisting pool of antibodies that could recognize common epitopes. Children show a different immune response to *S. pyogenes* than adults with lower levels of IgG3 (33). It would be interesting to study the antibody binding of young children to see if and when there is an elevation of basal *S. pyogenes* binding.

Nevertheless, generalizations based on our results should be made with care, since this study is based on a small study population. Still, the different infecting types in this study (*emm*1, *emm*85, and *emm*118) belong to three diverse *emm* groups, A-C, D, and E, respectively, and we added four more types (group A-C, *emm*4, *emm*5, and *emm*12; group E, *emm*89) (34), so it is reasonable to describe the opsonic response in our patients as broad. We believe that, taken together with previously published data, this study provides further proof of the development of a general rather than a strictly type-specific *S. pyogenes* immunity after infection.

## MATERIALS AND METHODS

**Patient inclusion and data collection.** Patients with *S. pyogenes* bacteremia in 2018 to 2020 were prospectively included in the study after oral and written consent were obtained. Acute-phase serum

was collected within 5 days after hospital admission, and convalescent-phase serum was collected after 4 to 6 weeks. Medical records of patients were reviewed to obtain clinical and epidemiological parameters. To our knowledge, none of the patients had a history of previous *S. pyogenes* infection during adulthood. The concentration of the immunoglobulins in serum was determined at the Department of Clinical Chemistry in Skåne, Sweden.

**Ethics.** The regional ethics committee of Lund University approved the study (2016/939, with amendment 2018/828).

**Sequencing.** The *S. pyogenes* isolates were collected from the blood of infected patients at the Laboratory for Clinical Microbiology, Lund University Hospital Sweden. Whole-genome sequencing was done at the Center for Translational Genomics at Lund University. NextSeq 550 Illumina sequencing was used to sequence the bacterial genomes. The genome sequencing data were searched against the CDC database of M protein families to detect the target M protein sequence. M protein sequences were pairwise aligned with the target M1 protein using the EMBOSS Needle web server. The four other isolates (types *emm*4, *emm*5, *emm*12, and *emm*89) used to test cross-reactivity were a gift through a collaboration with Oonagh Shannon's group, which has *emm* typed the isolates according to CDC protocol.

**Microbe strains.** The clinical isolates of *Streptococcus pyogenes* and the lab mutant of *S. pyogenes* SF370 with deficient M protein expression (dM) (35, 36) were statically cultured in Todd-Hewitt broth (Bacto) supplemented with 0.2% yeast extract (Difco) (THY) at 37°C and 5% $CO_2$. They were cultivated to log phase (optical density at 600 nm [$OD_{600}$], 0.3 to 0.4) (Ultrospec 10; Amersham Biosciences) before being heat killed at 80°C for 5 min. For the dM strain, group A antigen positivity was confirmed with a standard latex agglutination assay at the Laboratory for Clinical Microbiology, Lund University Hospital Sweden.

**Labeling and opsonization of bacteria.** Bacteria in phosphate-buffered saline (PBS) were first stained for 1 h at 37°C with 4 $\mu$M Oregon green 488-X, succinimidyl ester (Invitrogen), followed by 20 $\mu$g/mL CypHer5E (Cytiva) in $Na_2CO_3$ for the phagocytosis assay. For the supplementary phagocytosis assays, the stainings were combined and performed at 30 min with 75 $\mu$g/mL CypHer5E (Cytiva) instead. After staining, the bacteria were resuspended in Na medium (5.6 mM glucose, 127 mM NaCl, 10.8 mM KCl, 2.4 mM $KH_2PO_4$, 1.6 mM $MgSO_4$, 10 mM HEPES, 1.8 mM $CaCl_2$; pH adjusted to 7.3 with NaOH). To disperse any large aggregates, stained bacteria were then sonicated for 4 min (0.5 cycle, 75 A; VialTweeter) followed by determination of concentration using flow cytometry (CytoFlex; Beckman Coulter; lasers, 488 and 638 nm; filters, 525/40 and 660/10 nm).

Opsonization was performed on the experiment day with a bacterial concentration of 800,000 bacteria/$\mu$L at 37°C for 30 min with gentle shaking. For the supplementary phagocytosis assays, opsonization was performed less than a day earlier at a bacterial concentration of 500,000 bacteria/$\mu$L. Sera were heat inactivated before opsonization at 56°C for 30 min. The opsonins used, in addition to patient sera, were intravenous immunoglobulins pooled from healthy donors (IVIG; Octagam; Octapharma) and a humanized monoclonal IgG that is IgE specific (omalizumab [Xolair]; Novartis) and thus binds only to M protein via potential Fc binding.

**Binding assay.** Oregon green-stained bacteria were opsonized in a 1:2 serial dilution of sera, IVIG (2 mg/mL), and omalizumab (1 mg/mL) in a final volume of 10 $\mu$L. For the assessment across the different isolates, bacteria were double stained (Oregon green and CypHer5E) and opsonized with 5% sera or 0.5 mg/mL of omalizumab or IVIG. After opsonization, unbound antibodies were washed away three times by removing supernatant and resuspending the bacteria in 250 $\mu$L Na medium by centrifugation (3,000 $\times$ *g*, 5 min). Opsonized bacteria were stained with fluorescently labeled antibodies [Alexa Fluor 647-conjugated F(ab')2 fragment goat anti-human IgG Fab; Jackson ImmunoResearch Laboratories] at a 1:50 dilution for 30 min at 37°C. Data were obtained through CytoFlex, acquiring at least 15,000 events. Four separate bacterial colonies per isolate plate were picked and assessed.

**Affinity model.** Binding curves were analyzed using a GAS antibody binding model based on the transfer matrix method for competitive binding described in reference 25. The implementation of this model is available on GitHub (https://doi.org/10.5281/zenodo.4063760). Using this model, the binding of polyclonal antibody samples is characterized by the mean and range (95% confidence interval [CI]) of a log-normal distribution of affinities. The geometric means in the figures correspond to the antibody affinity of the polyclonal samples. Binding values were normalized to an interpolated saturation level before being evaluated with the model implementation. Measured binding curves are shown as the mean and standard deviation of data points, as described in the figure legends. Affinity values were derived by minimizing the weighted mean squared error of the model output and measured data using a MATLAB minimization function. The accuracy of predicted affinities was estimated using the bootstrap method, where the confidence intervals were calculated from 50 resamplings of the measured data.

**Determination of immunoglobulin subclasses.** Mass spectrometry analysis was performed on patient sera to measure the immunoglobulin subclasses. The sample preparation for mass spectrometry is described elsewhere (37). In brief, 8 M urea–100 mM ammonium bicarbonate was added to 1 $\mu$L of patient serum for denaturation, and 5 mM Tris(2-carboxyethyl)phosphine hydrochloride (TCEP) was added and incubated for 1 h at 37°C for reduction, followed by incubation with 10 mM iodoacetamide for alkylation at room temperature for 30 min. Samples were diluted in 100 mM ammonium bicarbonate and incubated overnight with 0.5 $\mu$g/$\mu$L sequencing-grade trypsin (Promega) at 37°C, followed by the addition of 10% formic acid-inactivated trypsin. SOLA$\mu$ horseradish peroxidase (HRP) at 2 mg/1-mL 96-well plate (Thermo Scientific) was used to concentrate the peptides (according to the manufacturer's instructions). The concentrated peptides were injected into a Q Exactive HF-X instrument (Thermo Scientific) connected to an Easy-nLC 1200 instrument (Thermo Scientific).

The peptides were analyzed in data-dependent acquisition mass spectrometry (DDA-MS) mode (37). In short, the peptides were separated on a 50-cm Easy-Spray column (column temperature 45°C;

Thermo Scientific) at a maximum pressure of $8 \times 10^7$ Pa with a linear gradient of 4% to 45% acetonitrile in 0.1% formic acid for 65 min. One MS full scan (resolution of 60,000 for $m/z$ 390 to 1,210) was performed, followed by MS/MS scans (resolution of 15,000) for the 15 most abundant ion signals. Precursor ions with a 2 $m/z$ isolation width were fragmented at a normalized collision energy of 30 using higher-energy collisional-induced dissociation (HCD). The automatic gain controls for the full MS scan were set to 3e6 and 1e5 for MS/MS. The DDA data were analyzed in MaxQuant (1.6.10.43) against a database comprising *Homo sapiens* (UniProt proteome identifier UP000005640), common contaminants from other species, and iRT peptides (38). For the search, tryptic digestion with a maximum of two missed cleavage was allowed. Carbamidomethylation (C) was set to static modifications, while oxidation (M) was set to variable modifications.

**Cell lines and primary cells.** The human monocytic cell line THP-1 (39) (TIB-202, male; American Type Culture Collection) was cultured in RPMI 1640 medium (Sigma-Aldrich) supplemented with 10% fetal bovine serum (FBS) (Life Technologies) and 2 mM GlutaMAX (Life Technologies) at 37°C in 5% $CO_2$. The cell density was kept between $0.2 \times 10^6$ and $1.0 \times 10^6$ cells/mL with viability over 95% (determined with erythrosin B [Sigma-Aldrich]), and cells were harvested at $0.5 \times 10^6$ to $0.8 \times 10^6$/mL for the phagocytosis assay without any differentiation. Neutrophil isolation was performed at room temperature with Polymorphprep (Abbott) according to the manufacturer's protocol on heparinized blood drawn from two different healthy donors.

**Phagocytosis assay.** The phagocytosis assay was performed and analyzed using the PAN method as described elsewhere (26). Briefly, phagocytosis was performed with 100,000 THP-1 cells or neutrophils in a final volume of 150 $\mu$L with different multiplicities of prey (MOP) from 0 to 400 at 37°C for 30 min with gentle shaking. The bacteria had, as previously described, been fluorescently double-stained and opsonized in either 5% sera or 0.1 mg/mL omalizumab. For the assessment across the different isolates, phagocytosis was performed at a MOP of 80, and the concentration of omalizumab and IVIG was 0.5 mg/mL. Phagocytosis was halted by transferring samples to ice, and samples were kept cold during data acquisition using CytoFlex. At least 5,000 events of the population of interest were acquired. Ice samples were used as a control for internalization. Free bacteria were analyzed separately to determine the fluorescent intensity of a single bacterial unit, and to confirm pH sensitivity of the staining, pH was decreased by adding 1 $\mu$L of sodium acetate (3 M, pH 5.0). Four different colonies per isolate were assessed for results shown in the figures, and two were used for the supplemental figures. For the results in the supplemental material, phagocytosis was performed at a MOP of 50 with a minimum of 1,000 events acquired.

**Analysis of flow cytometry data.** Flow cytometry data were analyzed using FlowJo version 10.6.2 (TreeStar). THP-1 cells were gated on forward scatter height (FSC-H) and side scatter height (SSC-H). Events with extreme negative fluorescence were excluded. Phagocytes positive for Oregon green (fluorescein isothiocyanate height [FITC-H]) were defined as associating, and those also positive for CypHer5E (allophycocyanin height [APC-H]) were defined as internalizing cells as well. Free bacteria were gated on SSC-H in combination with a positive Oregon green signal, and doublets were excluded by gating on FSC-H versus forward scatter area (FSC-A). Gating strategy is visualized in Fig. S2. The neutrophil gating was performed with same strategy after slightly widening the THP-1 cell gate at FSC-H versus SSC-H.

The data were then further analyzed using the PAN method (26) in Prism 9.3.1 (GraphPad Software). The built-in nonlinear analysis tool "Agonist versus response – Variable slope (four parameters)" was used to create curves and determine $MOP_{50}$, corresponding to the MOP which evoked half of the maximal response. One curve was generated per replicate, and means are presented in the figures. The persistent association was defined as the percentage of THP-1 cells positive for at least one bacterium either adhered to or internalized. In comparison, internalization was defined as THP-1 cells positive for at least one internalized bacterium. Normalization for association was performed by interpolating the median fluorescence intensity (MFI) at $MOP_{50}$.

To assess individual phagocyte ability, the amount of bacteria associated and internalized was determined by the MFI of an associated THP-1 cell. To convert it into the number of prey units per phagocyte (PxP), each fluorescent signal (associated, Oregon green; internalized, CypHer5E) was divided by the MFI of free bacteria for CypHer5E at pH 5. Since streptococci are typically not present as a single bacterium, a prey unit most likely represents a chain.

**Statistics.** To compare the acute- and convalescent-phase-serum effect on binding and phagocytosis of the corresponding isolate, the paired nonparametric Wilcoxon matched-pairs signed-rank test was used. No paring was performed if it had been normalized against acute-phase sera, and the hypothetical value was set to 100. Two-way analysis of variance (ANOVA) with Šidák' s multiple-comparison test was applied when the sera were tested across the different isolates. The alpha value was set to 0.05. For the statistical tests, Prism (version 9.3.1; GraphPad Software) was used, while data collection and simple calculations were performed using Microsoft Excel 2021 (Microsoft Corporation).

## SUPPLEMENTAL MATERIAL

Supplemental material is available online only.

**SUPPLEMENTAL FILE 1**, PDF file, 1.9 MB.

## ACKNOWLEDGMENTS

We acknowledge the Department of Clinical Microbiology, Office for Medical Services, Region Skåne, Lund, Sweden. We also thank Sebastian Wrighton and Arman Izadi for their important technical contribution.

T.d.N. was funded by the Royal Physiographic Society. P.N. was funded by Vetenskapsrådet (2018-05795, 2020-01511) and Alfred Österlunds Foundation. P.N. and J.M. was funded by Knut and Alice Wallenberg Foundation.

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
