## [Reviewer comments · Microbiology Spectrum]

Microbiology Spectrum

Invasive streptococcal infection can lead to the generation of cross-strain opsonic antibodies

Therese de Neergaard, Anna Bläckberg, Hanna Ivarsson, Sofia Thomasson, Vibha Kumra Ahnlide, Sounak Chowdhury, Hamed Khakzad, Wael Bahnan, Johan Malmström, Magnus Rasmussen, and Pontus Nordenfelt

Corresponding Author(s): Pontus Nordenfelt, Lund University

Review Timeline:

Submission Date:	June 30, 2022
Editorial Decision:	July 22, 2022
Revision Received:	October 7, 2022
Accepted:	October 7, 2022

Editor: Christopher LaRock

Reviewer(s): Disclosure of reviewer identity is with reference to reviewer comments included in decision letter(s). The following individuals involved in review of your submission have agreed to reveal their identity: Helena Bergsten (Reviewer #1)

Transaction Report:

DOI: <https://doi.org/10.1128/spectrum.02486-22>

July 22, 2022

Dr. Pontus Nordenfelt
Lund University
Department of Clinical Sciences
BMC B14
Lund SE-221 00
Sweden

Re: Spectrum02486-22 (Invasive streptococcal infection can lead to the generation of cross-strain opsonic antibodies)

Dear Dr. Pontus Nordenfelt:

On the basis of recommendations from expert reviewers in the field, I have determined that your manuscript requires edits before acceptance. Both reviewers found the study to overall be well-conducted and of high interest to the field, but had points for clarification, needed controls, and had extremely helpful comments on figure formatting that will improve the overall presentation for readers. While you are revising the manuscript, please specifically address the comments copied below from Reviewer 1 and Reviewer 2. Keep in mind to avoid speculation beyond that which is specifically demonstrated, otherwise experiments are required to support claims. Please also carefully check grammar and other writing throughout, referring to journal guidelines with regards to all formatting requirements.

Link Not Available

Sincerely,

Christopher LaRock

Journals Department
Reviewer comments:

Reviewer #1 (Comments for the Author):

General

Excellent paper describing the humoral immunity after invasive infection by *S. pyogenes* in 4 patients. The authors have primarily used flow cytometry and self-developed binding assays to study the titer and function of antibodies developed after infection in relation to a phagocytic cell line. The study is technically robust and relevant controls for the experiments have been included. Interesting data was generated, including a few surprising findings. The study verified that antibody binding is increased to the infecting isolate after an invasive *S. pyogenes* infection regardless of emm type. The largest surprises were that the patients infected with emm1-strains did not develop isolate-specific immunity and that opsonic activity of *S. pyogenes* antibodies is M-unspecific at high concentrations.

However, in the authors effort to further technical assessment of antibody function, a few aspects need to be assessed: 1) is phagocytosis of live bacteria same as that of heat-killed bacteria? 2) do cell-line monocytes phagocytose as well as primary phagocytes?

Technical concerns

-Live bacteria are different from heat-killed in several ways. Live bacteria produce PAMPs and toxins that stimulate inflammation and phagocytosis. However, live bacteria also have means of evading opsonophagocytosis. Also, bacteria have reasons to exist intracellularly, and heat-killed bacteria may not be able to internalize into phagocytes as well as live bacteria. Due to these differences, the functional efficacy of the sera in the study should be tested also on live bacteria.

-THP-1 cells are a simplified model of human monocytes and verification using freshly isolated peripheral-blood phagocytes is important. THP-1 expression of cell surface molecules and production of cytokines is different from that of primary cells, as well as their function. Also, primary cells have natural heterogeneity that cell lines do not. Hence, results should be verified using primary phagocytes.

Recommended additional experimental data:

-Test bacterial survival of the 4 isolates in a whole blood assay (that also includes other important immune cells such as neutrophils) with addition of matching patient sera (acute & convalescent). Assess colony forming units (CFU).

-Proof of group A antigen positivity and M-protein negativity of the M protein mutant strain (dM).

Additional questions/speculative thoughts:

-Were all isolates found in blood? Please specify.

-A modest increase in IgG levels in convalescence was identified, as well as a larger increase in bacterial binding, could this be explained by affinity maturation during infection? Please discuss.

-Antibody binding of isolates by convalescent AND acute sera is remarkably better than IVIG in Fig 2A-B for 3/4 patients. Is it possible that no patients were naïve to the infecting M-type and that adaptive immunity had kicked in with high production of specific antibodies even at collection of acute samples? That could explain the minor difference between acute and convalescent sera?

-Is there a correlation between the IgG increase with days of hospitalization? Patient cohort is very small, but one could speculate.

- Would this study have benefited of *S. pyogenes* infected children as controls? Since the acute sera is similar to IVIG in several assays, one wonders how this would look in immunologically naïve children.

Figures:

Fig 2A legend: I initially misread the graph and thought the authors had tested the all sera against isolate A-D here. Could legend be made more clear?

Fig 2E: Change {copyright, serif} to (E) .

Reviewer #2 (Comments for the Author):

See attached file

Staff Comments:

Preparing Revision Guidelines

- Point-by-point responses to the issues raised by the reviewers in a file named "Response to Reviewers," NOT IN YOUR COVER LETTER.

- Upload a compare copy of the manuscript (without figures) as a "Marked-Up Manuscript" file.
- Each figure must be uploaded as a separate file, and any multipanel figures must be assembled into one file.
- Manuscript: A .DOC version of the revised manuscript
- Figures: Editable, high-resolution, individual figure files are required at revision, TIFF or EPS files are preferred

Please return the manuscript within 60 days; if you cannot complete the modification within this time period, please contact me. If you do not wish to modify the manuscript and prefer to submit it to another journal, please notify me of your decision immediately so that the manuscript may be formally withdrawn from consideration by Microbiology Spectrum.

Comments and Suggestions for the Author:

General

Excellent paper describing the humoral immunity after invasive infection by *S. pyogenes* in 4 patients. The authors have primarily used flow cytometry and self-developed binding assays to study the titer and function of antibodies developed after infection in relation to a phagocytic cell line. The study is technically robust and relevant controls for the experiments have been included. Interesting data was generated, including a few surprising findings. The study verified that antibody binding is increased to the infecting isolate after an invasive *S. pyogenes* infection regardless of *emm* type. The largest surprises were that the patients infected with *emm1*-strains did not develop isolate-specific immunity and that opsonic activity of *S. pyogenes* antibodies is M-unspecific at high concentrations.

However, in the authors effort to further technical assessment of antibody function, a few aspects need to be assessed: 1) is phagocytosis of live bacteria same as that of heat-killed bacteria? 2) do cell-line monocytes phagocytose as well as primary phagocytes?

Technical concerns

-Live bacteria are different from heat-killed in several ways. Live bacteria produce PAMPs and toxins that stimulate inflammation and phagocytosis. However, live bacteria also have means of evading opsonophagocytosis. Also, bacteria have reasons to exist intracellularly, and heat-killed bacteria may not be able to internalize into phagocytes as well as live bacteria. Due to these differences, the functional efficacy of the sera in the study should be tested also on live bacteria.

-THP-1 cells are a simplified model of human monocytes and verification using freshly isolated peripheral-blood phagocytes is important. THP-1 expression of cell surface molecules and production of cytokines is different from that of primary cells, as well as their function. Also, primary cells have natural heterogeneity that cell lines do not. Hence, results should be verified using primary phagocytes.

Recommended additional experimental data:

- Test bacterial survival of the 4 isolates in a whole blood assay (that also includes other important immune cells such as neutrophils) with addition of matching patient sera (acute & convalescent). Assess colony forming units (CFU).
- Proof of group A antigen positivity and M-protein negativity of the M protein mutant strain (dM).

Additional questions/speculative thoughts:

- Were all isolates found in blood? Please specify.
- A modest increase in IgG levels in convalescence was identified, as well as a larger increase in bacterial binding, could this be explained by affinity maturation during infection? Please discuss.
- Antibody binding of isolates by convalescent AND acute sera is remarkably better than IVIG in Fig 2A-B for ¾ patients. Is it possible that no patients were naïve to the infecting M-type and that adaptive immunity had kicked in with high production of specific antibodies even at collection of acute samples? That could explain the minor difference between acute and convalescent sera?
- Is there a correlation between the IgG increase with days of hospitalization? Patient cohort is very small, but one could speculate.
- Would this study have benefited of *S. pyogenes* infected children as controls? Since the acute sera is similar to IVIG in several assays, one wonders how this would look in immunologically naïve children.

Figures:

Fig 2A legend: I initially misread the graph and thought the authors had tested the all sera against isolate A-D here. Could legend be made more clear?

Fig 2E: Change © to (E) .

Dear Editor and Reviewers,

We are grateful for the constructive comments we have received on our manuscript. We believe we have addressed all the identified concerns with either text modifications or new experimental data. Please see our detailed response to each of the points brought up by the reviewers. Line numbers refer to the main manuscript file and not the tracked changes file, where the line numbering is inconsistent.

Best regards,

Therese de Neergaard and Pontus Nordenfelt on behalf of the authors.

Reviewer 1

General

Excellent paper describing the humoral immunity after invasive infection by *S. pyogenes* in 4 patients. The authors have primarily used flow cytometry and self-developed binding assays to study the titer and function of antibodies developed after infection in relation to a phagocytic cell line. The study is technically robust and relevant controls for the experiments have been included. Interesting data was generated, including a few surprising findings. The study verified that antibody binding is increased to the infecting isolate after an invasive *S. pyogenes* infection regardless of emm type. The largest surprises were that the patients infected with emm1-strains did not develop isolate-specific immunity and that opsonic activity of *S. pyogenes* antibodies is M-unspecific at high concentrations.

However, in the authors effort to further technical assessment of antibody function, a few aspects need to be assessed: 1) is phagocytosis of live bacteria same as that of heat-killed bacteria? 2) do cell-line monocytes phagocytose as well as primary phagocytes?

Technical concerns

-Live bacteria are different from heat-killed in several ways. Live bacteria produce PAMPs and toxins that stimulate inflammation and phagocytosis. However, live bacteria also have means of evading opsonophagocytosis. Also, bacteria have reasons to exist intracellularly, and heat-killed bacteria may not be able to internalize into phagocytes as well as live bacteria. Due to these differences, the functional efficacy of the sera in the study should be tested also on live bacteria.

We have now performed paired experiments with live as well as heat-killed bacteria side by side. We see no qualitative or quantitative difference between the two conditions. The data for these experiments can be seen in the new Supplementary Fig 3. We have also added this in the methods section, and the results section at lines 321-322 and 325-328.

-THP-1 cells are a simplified model of human monocytes and verification using freshly isolated peripheral-blood phagocytes is important. THP-1 expression of cell surface molecules and production of cytokines is different from that of primary cells, as well as their function. Also, primary cells have natural heterogeneity that cell lines do not. Hence, results should be verified using primary phagocytes.

We have performed paired experiments with THP-1 cells and primary neutrophils. We see some donor variation among the neutrophils, but in general, while neutrophils are better at phagocytosis of bacteria, the same qualitative results are valid. Based on this data, we see no reason to invalidate the use of THP-1 cells as a model for neutrophil phagocytosis. Please refer to the new Supplementary Fig. 4, methods section, and the results section at lines 328-333.

Recommended additional experimental data:

-Test bacterial survival of the 4 isolates in a whole blood assay (that also includes other important immune cells such as neutrophils) with addition of matching patient sera (acute & convalescent). Assess colony forming units (CFU).

We and others have had major problems with the robustness and reliability of the blood-killing assays for group A streptococci, despite the interesting data they could add. These types of experiments also require much more sera to perform than our phagocytosis assay, which would limit the analytic power. Linked to that point is that we simply do not have enough convalescent sera to perform these experiments even if we

had wanted to try. Thus, the manuscript describes results on opsonization and not whether or not phagocytosis leads to killing in blood.

-Proof of group A antigen positivity and M-protein negativity of the M protein mutant strain (dM).

We have now validated that both the SF370 strain, and the delta M mutant we use are group A Streptococcus. This Wt and dM pair have been previously validated by western blot to lack the M protein (Bahnan et al 2021; bioRxiv). We have added this information to the manuscript at lines 132-134.

Additional questions/speculative thoughts:

-Were all isolates found in blood? Please specify.

Yes, they were isolated from blood. We have added this to the manuscript at line 117.

-A modest increase in IgG levels in convalescence was identified, as well as a larger increase in bacterial binding, could this be explained by affinity maturation during infection? Please discuss.

The reviewer raises an interesting point. It is a possibility, of course, that the infection has enhanced the pre-existing anti-streptococcal response by inducing another round of affinity maturation. We have added this possibility to lines 454-466 in the discussion.

-Antibody binding of isolates by convalescent AND acute sera is remarkably better than IVIG in Fig 2A-B for 3/4 patients. Is it possible that no patients were naïve to the infecting M-type and that adaptive immunity had kicked in with high production of specific antibodies even at collection of acute samples? That could explain the minor difference between acute and convalescent sera?

This is possible indeed. Most people have already been exposed to GAS as children, so adaptive immunity could be very quick. That could mean naive samples aren't exactly naive. Our best attempt at correcting for this was to use similar levels of antibodies in both serum and IVIG. But we do not have any control on how 'naive' the naive patients really are. We have added some thought on this to the discussion at lines 454-466.

-Is there a correlation between the IgG increase with days of hospitalization? Patient cohort is very small, but one could speculate.

This is an interesting point. If one correlates the IgG titers with the duration of hospitalization, a correlation does become apparent. Higher IgG titers correlate with longer hospitalization. This is, however not indicative of disease severity as that have multiple variables which could affect it. We believe there are too few patients to add thoughts about this in the manuscript.

- Would this study have benefited of S. pyogenes infected children as controls? Since the acute sera is similar to IVIG in several assays, one wonders how this would look in immunologically naïve children.

The reviewer again brings up an interesting point. We have added a reference to this in the discussion, see lines 464-466.

Figures:

Fig 2A legend: I initially misread the graph and thought the authors had tested the all sera against isolate A-D here. Could legend be made more clear?

This has now been fixed.

Fig 2E: Change © to (E) .

This has now been fixed.

Summary: This manuscript presents data claiming the opsonic antibody response to invasive group A *Streptococcus* is not a strong response specific to the infecting strain, but instead it is a broad modest response to the species. Using serum from four patients with bacteremia, the authors quantify the role of antibodies in the phagocytic response, and how it was different in acute versus convalescent serum. In addition, the authors claim not all antibodies included in this response are opsonic, showing the importance of measuring functional activity. Taken together, these findings represent a novel piece of knowledge that offers a different perspective from assumed dogma and would be of interest to the field. There are issues provided below that should be addressed to improve the quality of the manuscript.

Major Issues:

- There are many claims made throughout the paper about the immune response being broad to GAS instead of strain specific. This reviewer has concerns about the ability to make such broad claims using both a small sample of patients and a small sample of strains. The concerns are heightened as the M1 strains provided seemed to have a reduced immune response compared to the M118 and M85. To confirm the broadness of these claims, the authors should use more strains of GAS to investigate the broadness of the opsonization response.

We have now performed experiments with four additional strains, adding more data on this aspect. Please refer to new Supp. Fig 5, methods section, and lines 403-406 and 469-471.

- The phagocytic response to heat killed strains of GAS isolated from the patients are compared to a strain deficient in M protein. The authors do not show the antibody response is interacting with the M protein. The manuscript would be improved if the authors provided data showing what the antibodies are interacting with from GAS, and in what quantity from the serum. This could be done through an ELISA.

While M protein, likely plays an important role in the phagocytic response, we believe that the data actually shows that there must be other surface proteins responsible for many of the antibody interactions. This is mentioned at lines 303-306 indicating some potential proteins that could be responsible for this in the discussion. A collaborator is currently undertaking a major Ig:surface proteome interaction project. The results from that study will hopefully identify which surface epitopes from GAS are mostly targeted by immunoglobulins. We, therefore, believe this to be outside the scope of this current work.

Minor Issues:

- Lines 28 – 29: There is no discussion about correlation to clinical outcome in the paper, which is required to make a claim like this

We have now added some points about this at lines 430-437.

- Lines 43-47: The introduction to the paper is short and lacking detail needed for a general scientific audience. The section introducing group A *Streptococcus* is especially lacking detail

We have now added more information in lines 41-50.

- Lines 91 – 97: Because prior infection can lead to different memory responses, whether the patients had a history of GAS infections should be noted.

This has now been clarified at lines 107-108.

- Line 192: The name of THP-1 should be confirmed and method to differentiate should be stated if done.

We have now clarified the origin of the cell line in the methods and that the THP-1 cells were not differentiated. Please refer to lines 208 and 213.

- Lines 309 – 380: The discussion of patient data was very hard to follow as there was no consistent order or table to reference.

This has now been fixed. Please refer to Table 1.

- Lines 383 – 388: The data from the serum of Patient D was noticeably different from the other patients throughout the paper. The reviewer would have liked some discussion about the implications this may have, and whether it was from the delay to hospitalization or another reason

We have now discussed this. Please refer to lines 430-437.

- Lines 392 – 394: There is not sufficient data or sources to support confirming or refuting a claim of immune deficiency. If the authors are going to discuss this topic additional information is required.

We have reformulated this to be clearer to what data we are referring to. Please see lines 473-478 and 485-492.

- Lines 397 – 399: The mechanisms of GAS influencing the immune system should be discussed in additional detail

We have expanded the discussion on GAS immunity. Please see lines 496-522.

- Line 455: © should be corrected to the appropriate figure label.

We have now fixed this.

- Figure 2:
 - o The legend is incomplete and does not identify which strains are which.

We have now fixed this.

- o It is unclear what the significance noted in 2A and 2B is referring to, and the data is difficult to interpret.

We have added an explanation in the figure legend to what the significant differences refer to. We have also modified the figure annotations to indicate better that the samples in A are also a comparison of acute to convalescent sera. See Figure 2, and lines 301-303 and Figure legend 2.

- o Line 285 refers to differences being detected at 0.1% serum, but no statistical significance is provided, and differences are not visible to the reviewer due to wide error bars.

We have modified the text to refer to the trend rather than statistical significance at this serum level. See lines 302-303.

- Figure 3:
 - o The presentation of these data made interpretation difficult

We have added a schematic and explanation to Figure 3. Please see panel 3A and figure legend.

- o It is unclear what the significance in 3A – 3C is referring to, and the lack of it for 3D – 3E is noticeable.

We are comparing significant fold changes between acute and convalescent samples. We have added information on the significance to the figure legend. We only did statistics on the full curves (A-C) to use the full power of the experimental data. In panels D and E, we are describing the situation at MOP₅₀.

• Figure 4:

o It should be made clear that E,F, and G refer to convalescent vs acute sera without stating in separate text **We have placed the convalescent vs acute indication above the E, F, G to make it clearer. Please see Figure 4.**

October 7, 2022

Dr. Pontus Nordenfelt
Lund University
Department of Clinical Sciences
BMC B14
Lund SE-221 00
Sweden

Re: Spectrum02486-22R1 (Invasive streptococcal infection can lead to the generation of cross-strain opsonic antibodies)

Dear Dr. Pontus Nordenfelt:

Your manuscript has been accepted, and I am forwarding it to the ASM Journals Department for publication. You will be notified when your proofs are ready to be viewed.

Sincerely,

Christopher LaRock
Editor, Microbiology Spectrum
